# Complications of Percutaneous Vertebroplasty: A Pictorial Review

**DOI:** 10.3390/medicina59091536

**Published:** 2023-08-25

**Authors:** Mislav Cavka, Domagoj Delimar, Robert Rezan, Tomislav Zigman, Kresimir Sasa Duric, Mislav Cimic, Ivo Dumic-Cule, Maja Prutki

**Affiliations:** 1Clinical Department of Diagnostic and Interventional Radiology, University Hospital Centre Zagreb, Kispaticeva 12, 10000 Zagreb, Croatia; mislav.cavka@yahoo.com (M.C.); rrezan@gmail.com (R.R.); maja.prutki@gmail.com (M.P.); 2Department of Orthopaedic Surgery, University Hospital Centre Zagreb, Kispaticeva 12, 10000 Zagreb, Croatia; domagoj.delimar@kbc-zagreb.hr; 3School of Medicine, University of Zagreb, Salata 3, 10000 Zagreb, Croatia; zigman.tomislav@gmail.com (T.Z.); kresimir.sasa.djuric@kbc-zagreb.hr (K.S.D.); cimicmislav@gmail.com (M.C.); 4Department of Surgery, University Hospital Centre Zagreb, Kispaticeva 12, 10000 Zagreb, Croatia; 5Department of Neurosurgery, University Hospital Centre Zagreb, Kispaticeva 12, 10000 Zagreb, Croatia; 6Department of Nursing, University North, 104 Brigade 3, 42000 Varazdin, Croatia

**Keywords:** vertebroplasty, complication, cement leakage, spondylodiscitis

## Abstract

Percutaneous vertebroplasty is a minimally invasive treatment technique for vertebral body compression fractures. The complications associated with this technique can be categorized into mild, moderate, and severe. Among these, the most prevalent complication is cement leakage, which may insert into the epidural, intradiscal, foraminal, and paravertebral regions, and even the venous system. The occurrence of a postprocedural infection carries a notable risk which is inherent to any percutaneous procedure. While the majority of these complications manifest without symptoms, they can potentially lead to severe outcomes. This review aims to consolidate the various complications linked to vertebroplasty, drawing from the experiences of a single medical center.

## 1. Introduction

Percutaneous vertebroplasty is a minimally invasive procedure wherein bone cement is percutaneously injected directly into a fractured vertebral body. While various types of bone cement are accessible, polymethyl methacrylate (PMMA) is the most commonly utilized and is considered a reliable stabilizing agent for vertebroplasty [1]. Some growth factors, such as bone morphogenetic proteins, have undergone preclinical testing for similar applications, suggesting a potential alternative to PMMA in the future [2,3,4]. Typically, this procedure is guided by fluoroscopy, although occasionally CT scans are employed for precise needle placement and post-injection monitoring.

Vertebral compression fractures arise from trauma or the weakening of the bone structure due to conditions like osteoporosis or neoplasia, which are furthermore associated with an increased morbidity and mortality [5,6]. The practice of vertebroplasty is widespread for treatments on the lumbar and thoracic spine. In contrast, cervical vertebroplasty is executed by more skilled practitioners due to the smaller size of vertebral bodies and pedicles. Percutaneous vertebroplasty is recommended for cases of osteoporotic vertebral body compression fractures that persistently exhibit symptoms despite nonsurgical interventions, as well as for pathological fractures attributed to osteolytic metastases, spinal myeloma lesions, or vascular neoplasms. Absolute contraindications include local infection and untreated hematogenous infection [1].

Complications arising from vertebroplasty are stratified according to three levels of severity. Within the realm of mild complications, there are instances of temporary exacerbation of pain and transient episodes of hypotension. Moving on to moderate complications, these encompass occurrences such as infections and the seepage of cement into foraminal, epidural, or dural space. Severe complications emerge when the extravasation of cement takes place within paravertebral veins, potentially leading to pulmonary embolism, cardiac perforation, cerebral embolism, or even fatality [7]. Factors such as cortical destruction, the presence of soft-tissue masses in the epidural region, lesions exhibiting heightened vascularity, and significant vertebral collapse collectively contribute to an elevated likelihood of encountering complications. As a result, the frequency of complications is notably higher in cases of neoplastic vertebral collapse as opposed to osteoporotic collapse [8].

In this review, based on experiences from our center, we aimed to provide useful information which included practical advice that would help primarily in preventing potential complications when performing percutaneous vertebroplasty.

## 2. Mild Complications

Uncommon complications of vertebroplasty include a temporary rise in pain and fever. These occurrences are attributed to an inflammatory response triggered by the heat generated during the polymerization of polymethyl methacrylate [9]. Typically, postprocedure pain is mainly linked to the fracture itself, procedural complications, or the development of a vertebral body fracture at an adjacent level. Pain existing prior to the procedure is more likely associated with alternate conditions, such as undiagnosed degenerative disk disease preceding the vertebroplasty [1]. The management of pain and fever can be effectively achieved through the use of analgesics and antipyretics [2,8].

Transient arterial hypotension is an infrequent complication with unclear pathogenic mechanisms, although some hypotheses involve potential toxicity, vasodilatation, or allergic effects of the cement or bone marrow micro-emboli. It showed a satisfactory response to supportive measures and is considered self-limiting [2,10].

Fractures of the ribs, along with potential linked radiculopathy and pneumothorax, may arise. The most significant vulnerability to intraprocedural rib fractures is observed in patients with advanced osteoporosis, where fractures develop due to downward pressure applied to the chest wall of the patient during needle insertion. Radiculopathy is generally treatable using conservative approaches involving analgesics and is typically anticipated to subside within a few months [11].

## 3. Extravasation of Cement

Leakage of cement is a relatively common outcome of vertebroplasty and is identified as a main contributor to complications [5,12]. This occurrence can transpire through a cortical opening into the intervertebral disc space or adjacent paravertebral soft tissues, through the paravertebral veins, via the basivertebral foramen, or along the needle channel. Factors that influence the flow of cement into or out of a vertebral body can be grouped into three categories: parameters associated with the bone and fracture, attributes of the cement itself, and aspects of the injection technique (including parameters like volume, velocity, pressure, and needle placement). The density and properties of the vertebral bone play a significant role in determining cement flow. Higher bone density and increased bone quality provide a more stable environment for cement placement and reduce the risk of leakage. Osteoporotic or severely weakened bone may have compromised structural integrity, and thereby an increased risk of cement extravasation. The type and configuration of the vertebral fracture can impact cement flow. For example, in cases of a simple compression fracture, cement tends to flow more readily into the fractured region. In complex fractures with multiple fracture lines or clefts, it may be challenging to achieve uniform cement distribution, leading to potential leakage [13]. The viscosity of the cement affects its flow characteristics. Therefore, higher viscosity may resist easy flow, requiring higher injection pressures and potentially increasing the risk of leakage. In contrast, lower viscosity cements may flow more easily but can also be associated with increased leakage risk if not controlled properly. The setting time of the cement determines the duration during which it remains injectable, while the mechanical properties of the cement, such as its elasticity and compressive strength, can impact its ability to stabilize the fractured vertebral body and resist dislodgment or fracture [14]. The volume of injected cement affects the distribution and potential leakage. Injecting an excessive volume of cement can increase the pressure within the vertebral body and lead to unwanted extravasation. The rate of cement injection and the applied pressure influence how well the cement fills the voids within the vertebral body. Thus, excessive pressure can force cement into unintended spaces, while slower injection may allow for more controlled distribution [15]. Some of these parameters have already been studied, although the interaction of cement with the structure of the vertebral body and details regarding how the cement extravasates from the vertebral body are still poorly understood [5].

The leakage of cement material into the spinal canal or neural foramen has been described as epidural leakage, a phenomenon that can manifest not only via defects in the posterior wall of the vertebrae or through the basivertebral foramina, but also through the anterior internal venous plexus. Certain observations suggest that the incidence of epidural leakage is most pronounced in the upper thoracic vertebral bodies, irrespective of the quantity of PMMA injected. When it comes to vertebroplasty procedures above the T-7 level, it is crucial that these are undertaken by a skilled and experienced specialist, given the smaller size of the vertebral bodies and pedicles [16]. The higher rate of cement extravasation is expected following administration of the larger amount of PMMA. Identifying leakage by fluoroscopy or X-ray is difficult and inter-observer agreement dependent. Therefore, a CT scan is the method of choice for precise assessment of the rate of cement extravasation and could help with detecting whether postoperative clinical symptoms are associated with leakage [12]. Most cement leakages are asymptomatic but can also cause compression and subsequently lead to a severe clinical consequence such as paraplegia, spinal cord compression, cement pulmonary embolism, and even death. Neurologic symptoms may be temporary, probably due to local inflammation, or may be caused by the direct compression by the cement [5,17]. Delayed onset of neurological symptoms, manifested as L4 radiculopathy following cement leakage, was recently reported [18]. 

### 3.1. Epidural and Foraminal Cement Leakage

Cement leakage into the foraminal or epidural spaces may lead to compromised neurological status due to spinal cord or nerve root damage, which is dependent on the volume of the leakage (Figure 1). Most cases remain clinically asymptomatic or with negligible symptoms [2].

Cement leakage into the epidural space can be classified into three distinct categories: type B entails leakage via the basivertebral vein, type S involves the segmental vein route, and type C arises through a cortical breach. Leaks of type B originate from the vascular foramen and advance into the spinal canal, disseminating along the epidural venous plexus. Type S leaks usually follow a horizontal trajectory, tracing alongside the segmental veins. On the other hand, type C leakage emerges from a cortical disruption encircling a vertebral body, potentially extending into the spinal canal (refer to Figure 2) [19].

### 3.2. New Vertebral Fractures

Leakage of cement into the intervertebral disc space during vertebroplasty increases the risk of subsequent fractures in neighboring vertebral bodies. Hence, it is recommended to position the needle laterally and away from the vertebra’s central axis, especially in cases of fractures located at the vertebra’s center. Moreover, adjustments to the cement’s viscosity and volume are advisable to enhance its consistency and reduce the likelihood of leakage [2]. It is vital to highlight that the literature lacks a prospective randomized study on the occurrence of new vertebral fractures in osteoporotic patients with vertebral collapses, comparing those treated with vertebroplasty with those treated with conservative management [7]. Some researchers have concluded that adjacent-level fractures subsequent to vertebroplasty are more likely attributable to underlying osteoporosis rather than the procedure itself [1].

### 3.3. Paravertebral Soft Tissue Leakage

Cement leakage into the paravertebral soft tissues typically lacks clinical significance. Such leakage might stem from preexisting cortical damage or from the cortical breach occurring during the biopsy preceding the vertebroplasty (see Figure 3) [7]. While instances are infrequent, there have been reports of temporary femoral neuropathy and instances requiring surgical removal of cement deposited within the paravertebral soft tissues [11].

### 3.4. Venous System Cement Leakage and Pulmonary Embolism

Leakage was more frequently observed within the perivertebral venous plexus in comparison to the nearby intervertebral discs or the surrounding soft tissues (see Figure 3). The venous network traversing the vertebral column consists of three primary interconnected systems: the internal venous plexus, the external venous plexus, and the basivertebral system. Originating within the anterior one-third of the vertebral body, the basivertebral veins converge towards the posterior region, where they drain into the anterior part of the internal venous plexus. In the anterior area, these basivertebral veins merge with the external plexus. Positioned on the dorsal surface of the vertebral body, the exit point of the basivertebral vein lies centrally between the pedicles. The anterior component of the internal venous plexus empties into the segmental veins, which exit the spinal canal through the foramen located between the nerve root and the medial aspect of the pedicles. This implies a direct venous link connecting the bone marrow with the foraminal space [20].

Three mechanisms are considered responsible for cement embolism following vertebroplasty: insufficient polymerization of the polymethylmethacrylate at the time of its injection, incorrect needle positioning at the time of cement injection, and overfilling of the vertebral body with cement, resulting in cement migration into the venous system [21]. If the cement is inadequately polymerized at the time of injection, it may remain partially liquid or have a semi-solid consistency. This increases the risk of leakage from the vertebral body, allowing cement to enter the surrounding blood vessels leading to potential complications due to emboli. It is important for medical professionals to ensure proper preparation and polymerization of the PMMA bone cement before injection. This may involve thorough mixing of the cement components, appropriate timing to allow for complete polymerization, and adherence to established procedural guidelines to minimize the chances of complications like cement embolisms [22].

The only sign that can predict the development of a pulmonary cement embolism is fluoroscopic evidence of cement leakage to the azygos vein or vena cava during vertebroplasty (Figure 4 and Figure 5). Bone cement extravasates toward the vertebral venous plexus which is connected to the azygos system by which it reaches the inferior vena cava, the right cardiac chamber, and finally the pulmonary arterial system which may lead to a potentially fatal pulmonary embolism [23,24].

A symptomatic pulmonary embolism following vertebroplasty can manifest through either the migration of cement or the migration of fat and bone marrow cells. The majority of instances involving radiologically identified PMMA migration into lung vessels demonstrate no symptoms. In this context, occurrences of fat tissue embolisms tend to surpass those of PMMA embolisms [25]. Clinical indications of a pulmonary cement embolism encompass the abrupt onset of dyspnea, tachypnea, tachycardia, cyanosis, chest pain, cough, hemoptysis, and perspiration following vertebroplasty. Despite the initial lack of symptoms in numerous cases, many cement emboli are fortuitously identified during subsequent imaging assessments [24]. It is imperative to initiate early detection and prompt management even in the absence of clinical signs. Should respiratory symptoms emerge post vertebroplasty, a meticulous evaluation for potential pulmonary cement embolism is warranted [23,24]. In situations where the cement advances into the right ventricle, yet proves too lengthy and rigid to traverse the pulmonary artery, it may lodge in the heart and give rise to cardiac perforation—a truly uncommon complication of vertebroplasty, with only a single case documented in the literature. This particular case was potentially fatal due to hemopericardium and tamponade [2,26]. Instances of cerebral embolus have also been reported, with reports attributing them to fat emboli originating from heightened intramedullary pressure during the cementation process [2,27].

## 4. Infection

While percutaneous vertebroplasty is a procedure known for its minimal invasiveness, there remains a risk of postprocedural infection, a concern inherent to any percutaneous intervention [28]. Reported infections encompass discitis, osteomyelitis, and potentially epidural infections. The etiology of spondylitis following vertebroplasty can be categorized into three primary groups: preexisting spondylitis, infection triggered by the procedure itself, and infection originating from hematogenous seeding. Infections occurring shortly after the procedure could arise from performing the intervention on an already infected vertebra, perhaps misdiagnosed as an osteoporotic fracture, particularly in cases of conditions like tuberculosis, or on fractured vertebrae coexisting with spondylitis. When an infection occurs within a two-month timeframe, preexisting spondylitis and procedure-induced infection should be considered. In scenarios where the interval between the procedure and the occurrence of infection is brief, the likelihood of preexisting spondylitis is higher than infection caused by the procedure (see Figure 6). Although hematogenous seeding-induced infection can also result in early infection, it predominantly leads to late-onset infections [29,30].

Prevention of infection following vertebroplasty includes preoperative routine checks of inflammatory parameters (C-reactive protein and white blood cell count) and MRI. Enhanced MRI should be performed if inflammatory markers are elevated with suspected infection. For patients with any type of acute infection, the procedure should be delayed until the infection improves and inflammatory parameters decrease. Therefore, a one- to two-weeks window period of conservative management is suggested before the cement augmentation procedure, to exclude infection in patients with an acute compression fracture and elevated inflammatory parameters [29,30].

In the case of immunocompromised patients with comorbidities, conservative treatment could lead to better outcomes than cement augmentation. For high-risk patients, some authors recommended the routine addition of tobramycin to the cement, while others suggested usage of perioperative intravenous prophylactic antibiotics. When infection occurred due to vertebroplasty/kyphoplasty surgical debridement, stabilization is the method of choice. However, a considerable number of studies reported that conservative treatment with antibiotics may cure an infection of PMMA in the vertebrae [29].

## 5. Reducing the Risk and Effects of Complications

In order to avoid complications, procedure technique should allow access through the transpedicular route in the lumbar spine and via the costovertebral junction in the thoracic spine, thereby avoiding a cortical breach when possible. Opacification of cement should be optimized by following the manufacturer recommendations, not exceeding recommended proportions of powder and liquid polymer, and defining the optimal cement viscosity before injection. If cement leakage occurs, termination of the procedure is recommended [11]. A preoperative check of inflammatory parameters and MRI should be mandatory. Persistent severe pain, new severe pain of a different character, signs of spinal canal stenosis, or sudden onset of respiratory symptoms should be sent to emergency diagnostics to exclude complications and start treatment if necessary [1,23,24].

Outpatient clinic follow-up should occur two to four weeks post procedure when the patient should again be assessed for signs of procedural complications. Subsequent follow-up appointments may take place if indicated [1].

## 6. Discussion

Vertebroplasty is a potentially life-changing procedure for individuals suffering from vertebral compression fractures, osteoporosis, neoplasia, and other bone-structure-weakening conditions. However, complications can arise from the procedure, ranging from mild and temporary side effects to more severe and potentially life-threatening ones. Mild complications may include a temporary increase in pain and hypotension. Moderate complications may involve infection and cement extravasation into the foraminal, epidural, or dural space. Lastly, severe complications include instances of cement extravasation into paravertebral veins and can result in pulmonary embolism, cardiac perforation, and cerebral embolism, which can be fatal [7]. Pulmonary cement embolism is dependent on a number of factors such as affected vertebrae, lesion localization, and puncture method, which are understood as separate risk factors. Surgeons should take these factors into account when devising treatment approaches [31].

## 7. Conclusions

It is important for clinicians to obtain a clear understanding of the patient’s condition before performing the procedure, as well as to be vigilant in following proper protocols during the procedure. Additionally, utilizing a combination of fluoroscopy and CT for needle positioning and injection assessment is also recommended. In conclusion, while vertebroplasty is an effective procedure for treating vertebral compression fractures, it is also important to be aware of the potential complications and take necessary precautions to reduce risks. Percutaneous vertebroplasty is considered a safe and effective option in the management of vertebral fractures. Although the majority of complications following percutaneous vertebroplasty are asymptomatic, serious complications can occur, which is why careful and precise techniques should be executed during the procedure to minimize the risk.

## Figures and Tables

**Figure 1 medicina-59-01536-f001:**
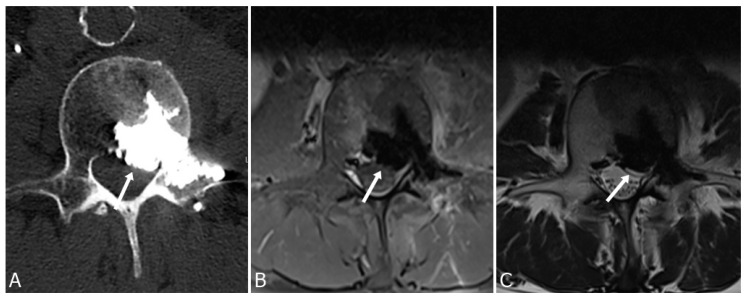
Cement leakage into epidural space with compression of the dural sac (arrows) seen on axial CT image (**A**), T1-weighted (**B**), and T2-weighted (**C**) MRI images, following vertebroplasty of L2 vertebrae in a 67-year-old man.

**Figure 2 medicina-59-01536-f002:**
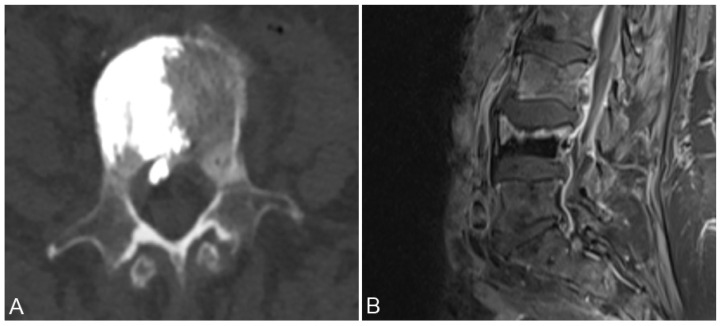
Axial CT image (**A**) and T2 weighted sagittal MRI image (**B**) show discrete intraspinal bone cement leakage.

**Figure 3 medicina-59-01536-f003:**
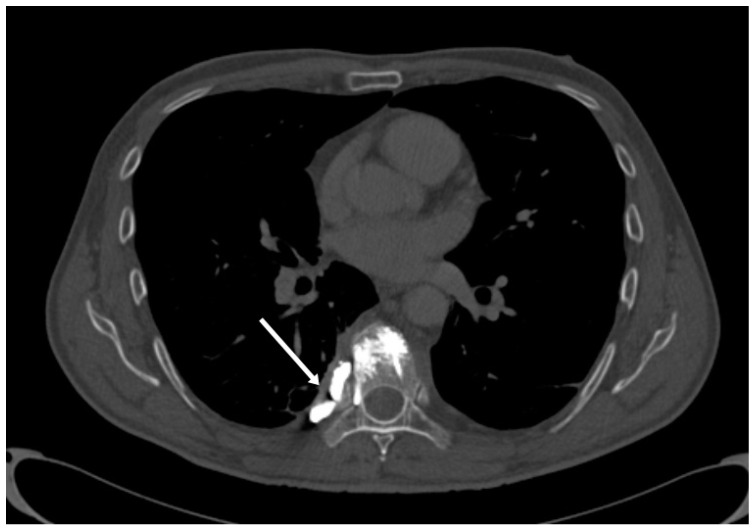
Cement leakage into the right subpleural space following vertebroplasty of Th8 vertebra (arrow).

**Figure 4 medicina-59-01536-f004:**
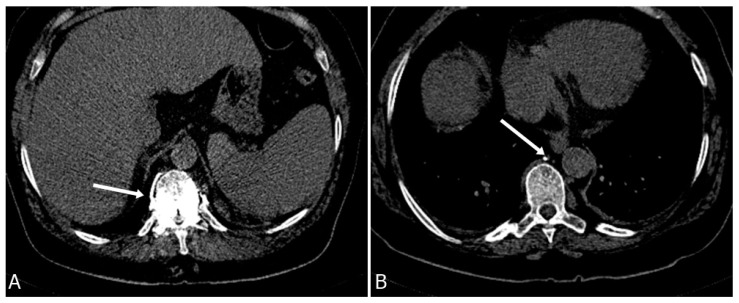
Axial CT scan of a 64-year-old woman following vertebroplasty of Th12 vertebrae with an arrow pointing to a cement leak into the paravertebral venous system (**A**). Cement can be seen in the azygos vein (**B**). Patient remained asymptomatic and late cement migration during follow-up did not occur.

**Figure 5 medicina-59-01536-f005:**
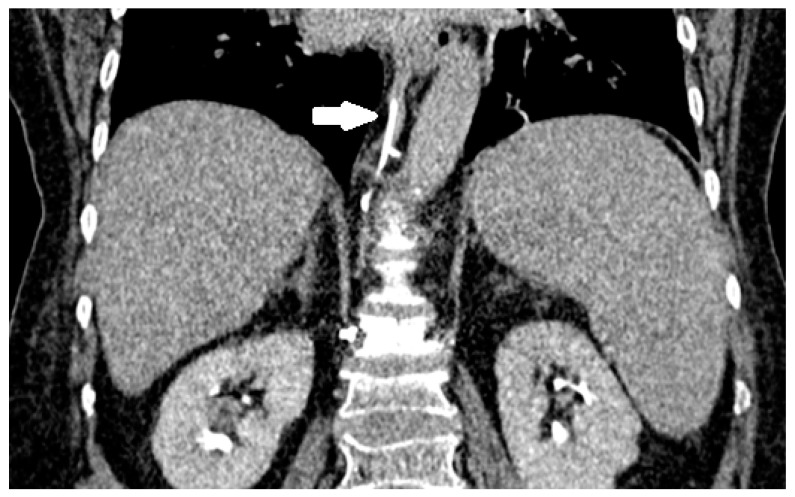
Sagittal CT scan of the same patient. Cement can be seen in the azygos vein (arrow).

**Figure 6 medicina-59-01536-f006:**
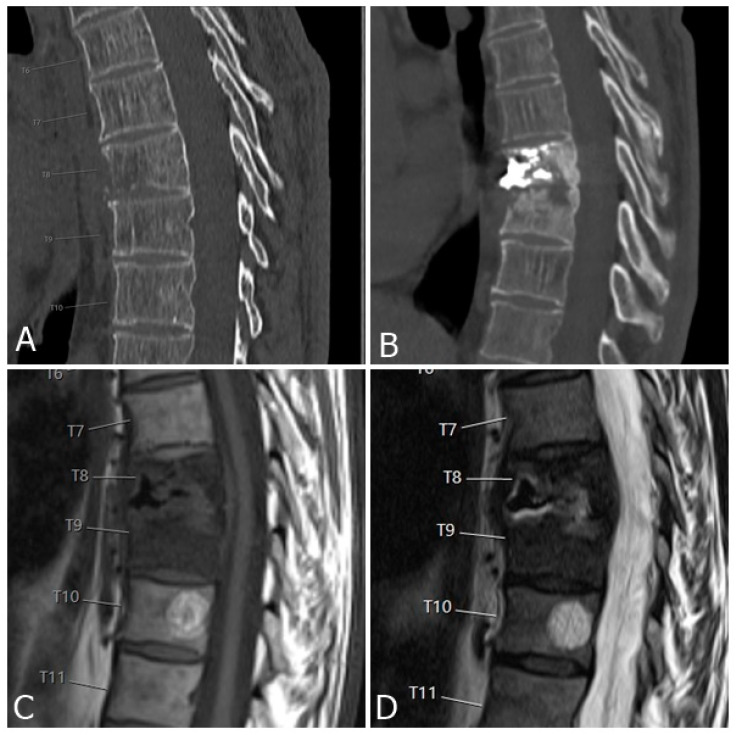
(**A**) CT scan of the thoracic spine of a 59-year-old man shows infiltration of the Th8 vertebral body by a neoplastic process. Biopsy and vertebroplasty were performed. Three months after procedure, the patient was presenting with new severe back pain and high C-reactive protein tests. CT shows the loss of intervertebral disc space height and bony destruction of Th8 and Th9 vertebrae (**B**). Bony destruction and low signal in disc space and adjacent endplates can be seen on T1-weighted images (**C**), as well as high signal in disc space and adjacent endplates on T2-weighted images (**D**), consistent with fluid in disc space and bone marrow oedema, all suggesting advanced spondylodiscitis.

## Data Availability

Data used for analysis are contained within the article.

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
