# Peer review of "Complications of Percutaneous Vertebroplasty: A Pictorial Review"

_medicina, 2023, doi:10.3390/medicina59091536_

Round 1

Reviewer 1 Report

This review describes the common and less common complications of vertebroplasty. The introduction is well written. The guided review through the complications is well written.

Several points i would consider when re-writing the article:

1. There is almost no mentioning  of kyphoplasty which is a much more common procedure in certain parts of the world. As a guided review i would prefer to see also the details about kyphoplasty and the difference in complication rates. It would add a great benefit to the article and to the readers, and will increase the interest in reading it.

2. I would like to see references of clinical trials and studies rather than other reviews. I think that the the reference list lacks (although contain some) up-to-date studies.

3. consider add some information regarding conservative treatment as it is without any surgical complications. consider adding more information about the treatment decision algorithm in your institution. 

Author Response

We thank this reviewer for the constructive criticism which has improved the quality of our revised manuscript.

1) Thank you for this constructive suggestion. However, in our review we planned to explain details about percutaneous vertebroplasty and no kyphoplasty since we perform it in our center and therefore believe that are competent enough to provide a comprehensive review focused on percutaneous vertebroplasty that would be useful for everyday clinical practice.

2) Thank you for this remark. In the revised version of the manuscript, we added several important and relevant clinical trials for this procedure. Thus, we highly appreciate your suggestion since it markedly increased the quality of our review. Added references are marked with red: Grgurevic L et al, Carli D et al, Gu YF et al.

3) Although we know that conservative treatment is of interest for particular specialists, we were focused on the procedures we perform on the Department of Interventional Radiology. Therefore, we aimed to provide practical advices that would help in performing procedures and preventing potential complications.

Reviewer 2 Report

General

The aim of this study was to find efficacy of complications of percutaneous vertebroplasty

I really agree your concept and well written manuscript.

I am positive in terms of publication but still have several question for correction.

I have the following detailed comments

1.       Introduction

Well written. But as possible, It seems necessary to clarify the purpose of writing this article.

2.     Mild complication and possible rib fractures

Well written. It is better to include the term possible rib fractures in the title in Mild complication. Or if you think possible rib fracture is important, I recommend separating the sections.

3.     Cement extravasation

Well written. The article briefly explained the factors for cement flow (Line77-80), but it would be better to explain the factors separately in detail. Otherwise, it is considered difficult for readers to understand this paper unless they have a high level of background knowledge.

Line 95-97 seems to be a confusing sentence. It would be better to separate the content that the spinal cord is directly compressed and the content that temporary symptoms occur, as it can cause confusion to readers.

You explain 3 types of cement leakage in line 108-113. It would be better to add example figure of each subtype.

Duplicate numbering in Line 99 and 119. And also in Line 117 and 138 It can be a fatal mistake and needs to be fixed.

You need to write in a little more detail how insufficient polymerization affects cement embolism. (Line 152-155)

In figure 4, for clear understanding, the location of lesion should be marked with an arrow or the like. And looking at what you have explained, figure 4 is said to be “during vertebroplasty”, but in fact I think it is “postoperative evaluation”. Content correction or picture correction seems necessary.

4.     Infection

I think the contents of Figure 5 should also be present in the text.

5.     Reducing the risk and effects of complication

The word “avoid” is not fit in sentence (Line 235).

6.     Discussion

Well written. At the end, as a result of the research of this study, it seems good to suggest a routine test or protocol after surgery.

It would be better to separate conclusion into separate sections (Line 257-263).

It would be better to recheck english language. Because of a few awkward word, I'm worried that this article's value could be evaluated as less valuable.

Author Response

We thank this reviewer for the constructive criticism which has raised important questions and significantly improved the quality of our revised manuscript. Reviewers′ comments are marked in text with red color.

1) At the end of introduction paragraph, we added an explanation of primarily purpose of this review.

2) According to your suggestion, we removed the term “possible rib fractures” from the paragraph title.

3) In revised version of the manuscript, we provided detailed explanation about factors that influence cement flow. New references are added as well.

4) We clearly separate the asymptomatic leakage from compression due to leakage to avoid any misunderstanding. Changes are written in red letters.

5) We corrected those mistakes and would like to thank to reviewer for avoiding potential misleading mistakes. Figure numbers are corrected as well in the new version of the manuscript.

6) Thank you for this valuable advice. Polymerization is described in details and changes are marked with red.

7) According to your suggestion arrow is inserted in Figure 4, pointed leakage in azygos vein.

8) Figure 5 explanation is inserted in the text.

9) We correct as it was suggested and now sentence is very easy to understand. New construction is marked with red color.

10) Conclusion is separated in new paragraph as suggested.

Round 2

Reviewer 2 Report

Thanks for well revised article.

None.